# Planar photonic chips with tailored angular transmission for high-contrast-imaging devices

Yan Kuai [1], Junxue Chen [2], Zetao Fan [1], Gang Zou [3], Joseph. R. Lakowicz [4] & Douguo Zhang [1]✉

A limitation of standard brightfield microscopy is its low contrast images, especially for thin specimens of weak absorption, and biological species with refractive indices very close in value to that of their surroundings. We demonstrate, using a planar photonic chip with tailored angular transmission as the sample substrate, a standard brightfield microscopy can provide both darkfield and total internal reflection (TIR) microscopy images with one experimental configuration. The image contrast is enhanced without altering the specimens and the microscope configurations. This planar chip consists of several multilayer sections with designed photonic band gaps and a central region with dielectric nanoparticles, which does not require top-down nanofabrication and can be fabricated in a larger scale. The photonic chip eliminates the need for a bulky condenser or special objective to realize darkfield or TIR illumination. Thus, it can work as a miniaturized high-contrast-imaging device for the developments of versatile and compact microscopes.

[1] Hefei National Laboratory for Physical Sciences at the Microscale, Advanced Laser Technology Laboratory of Anhui Province, Department of Optics and Optical Engineering, University of Science and Technology of China, 230026 Hefei, Anhui, China. [2] College of Science, Guilin University of Technology, 541004 Guilin, Guangxi, China. [3] CAS Key Laboratory of Soft Matter Chemistry, Department of Polymer Science and Engineering, University of Science and Technology of China, 230026 Hefei, Anhui, China. [4] Center for Fluorescence Spectroscopy, Department of Biochemistry and Molecular Biology, University of Maryland School of Medicine, Baltimore, MD 21201, USA. ✉email: dgzhang@ustc.edu.cn

Modern microscopes can produce images of high resolution and high magnifications[1–3]. Enough contrast of the image is also essential to clearly reveal the details of the specimens[4,5], resulting in the widespread use of fluorescent probes. A variety of techniques have been developed to improve image contrast without modification of the samples (label-free imaging), such as phase contrast imaging, differential interference contrast (DIC), and Hoffman modulation contrast. These contrast enhancing techniques require specialized and expensive additional components. For examples, the equipment needed for DIC microscopy includes a polarizer, a beam-splitting modified Wollaston prism below the condenser, a beam-recombining modified Wollaston prism above the objective, and an analyzer above this upper prism. The phase contrast and Hoffman modulation contrast techniques need a specialized condenser and objective[1]. These specialized techniques need additional optical or mechanical components, thus complicating the configuration of the microscope and increasing the complexities in operations. Another widely used approach is darkfield illumination, which is particularly suitable for specimens that display little or no absorption and/or weakly absorbing biological samples. Darkfield microscopy (DFM) has been widely used in many fields of science and engineering, such as biological imaging, nanoparticle characterization and inspection of semiconductor devices[6–9]. However, it also cannot be a simple and inexpensive imaging system. In a typical DFM, first, the specimen is illuminated at oblique angles far from the direction normal to the sample, then a bulky darkfield condenser is needed which collects light at high angles above the critical angles. Second, only light that is scattered by the specimen into a cone of apex angle cantered around the microscope's optical axis. To meet this requirement, the objective is chosen such that it collects rays over a small range of angles which are far from the normal axis, so no light directly from the darkfield condenser contributes to the image. The regions on the specimen where there are no small features to scatter light are almost completely dark, often resulting in high-contrast images and giving "darkfield" microscopy its name. Third, the specialized condenser, objective and additional components are prone to misalignment and add cost and complexity to the microscope and decrease opportunities for small size devices for imaging[10,11]. The use of a bulky condenser also results in the very small illumination area (of micrometer scale).

In recent years, there has been interest in development of new imaging instruments with the nanophotonic structures to downsize or simplify the microscope set-up, improve the imaging performance, and decrease complexity. For example, two multifunctional and compact metasurface layers were used to develop a compact phase gradient microscope, which can generate a quantitative phase gradient image with increased image contrast[12]. The combination of ptychographic coherent diffractive imaging with sub-surface nanoaperture arrays was shown to yield an enhancement of both the reconstructed phase and amplitude images[13]. A luminescent photonic substrate with a controlled angular fluorescence emission profile was used in a conventional microscopy to replace the bulk condenser for miniaturized lab-on-chip darkfield imaging devices[14,15].

We demonstrate that after the attachment of a planar photonic chip to the substrate of a standard brightfield microscopy (BFM), both darkfield and total internal reflection (TIR) imaging can be realized in one experimental set-up without the use of a bulky darkfield condenser (DFC) and other specialized components. The new microscopes can be named as chip-based darkfield microscopy (C-DFM) and chip-based total internal reflection microscopy (C-TIRM). The C-DFM and C-TIRM have the merits of large illumination area, high imaging contrast, simple configuration and easy for optical-alignment. Both DFM and TIRM emphasize the high-spatial-frequency components associated with small features in the specimen morphology and in some imaging scenarios, can even provide resolution beyond the diffraction limit[16,17]. Different from the DFM that uses far-field propagating light as the illumination source, the TIRM uses pure evanescent waves on the surface as the illumination source, which will have higher spatial frequency and are more sensitive to the changes on the surface. It is ideally suited to analyze the localization and dynamics of molecules and events occurring near the interface, such as the plasma membranes and surface-bound single molecules.

## Results

**Configuration of the planar photonic chip.** The proposed chip is designed to provide evanescent wave excitation (or TIR) at 640 nm wavelength and darkfield conditions at 750 nm wavelength, using a standard brightfield microscope. The photonic chip consists of three parts (Fig. 1a). The middle is a dielectric layer (thickness about 2 μm) doped with $TiO_2$ nanoparticles (diameter at 60 nm). The bottom and top are the dielectric multilayers with different photonic band gaps (PBGs)[18,19]. The multilayers are made of alternating $SiO_2$ and $SiN_x$ layers. Details of the structural parameters are given in Fig. S1. The color scale encoded reflectivity (Fig. 1b, c) of the bottom and top multilayer was calculated by using transfer matrix method[20].

For the bottom multilayer, when the incident beams are of transverse-magnetic (TM) or transverse-electric (TE) polarization, there are reflection valleys at nearly 0° (Fig. 1d), meaning that only a near-normal incident beam can transmit through this multilayer (inset of Fig. 1d). The unusual transmission of the bottom multilayer layer is the result of two periodical dielectric structures, separated by a thicker layer of $SiO_2$ (Fig. S1). Surface normal transmission occurs at designed wavelengths (640 and 750 nm here) and no transmission happens at other angles for the designed wavelengths (Fig. S2). This restricted surface-normal transmission prevents the scattered light from leaving the bottom layer.

When the transmitted beam reaches the middle layer containing $TiO_2$ nanoparticles, its propagating direction will be changed due to the scattering by the nanoparticles. A portion of the scattered light returns to the bottom multilayer but will bounce back to the scattering layer due to the PBG of the bottom multilayer, and only scattered light at near normal angle can escape through the bottom multilayer. Some of the scattered light will reach the top multilayer. The PBG of top multilayer was designed (Fig. 1c) so that the scattered light at 750 nm wavelength can transmit at designed polar angles (from 27° to 42° in glass, within numerical aperture (NA) < 0.7, the reflection valleys on Fig. 1e, TM polarization). For the scattered light with propagating directions within NA < 0.7, it will be reflected to the scattering layer, and then reflected by the bottom multilayer. When the NA of the imaging objective is <0.7, darkfield imaging can be realized[1,21]. Light scattered at 640 nm wavelength lies in the forbidden band (Fig. 1c). When the scattering angle is larger than the critical angle, TIR and evanescent waves will occur at the multilayer/air interface, resulting in TIR imaging[11].

In brief, the scattered layer provides light of various propagating directions and thereby replaces the bulky DFC. The role of the top multilayer is to select the transmitting angles of the scattered light, either larger than a designed polar angle or induced evanescent waves on the top surface. The bottom multilayer only allows the transmittance of near-normal incident beams, so it can recycle scattered light into propagation angle ranges that are transmitted by the top multilayer, and then enhance the light intensity on or out of the top multilayer.

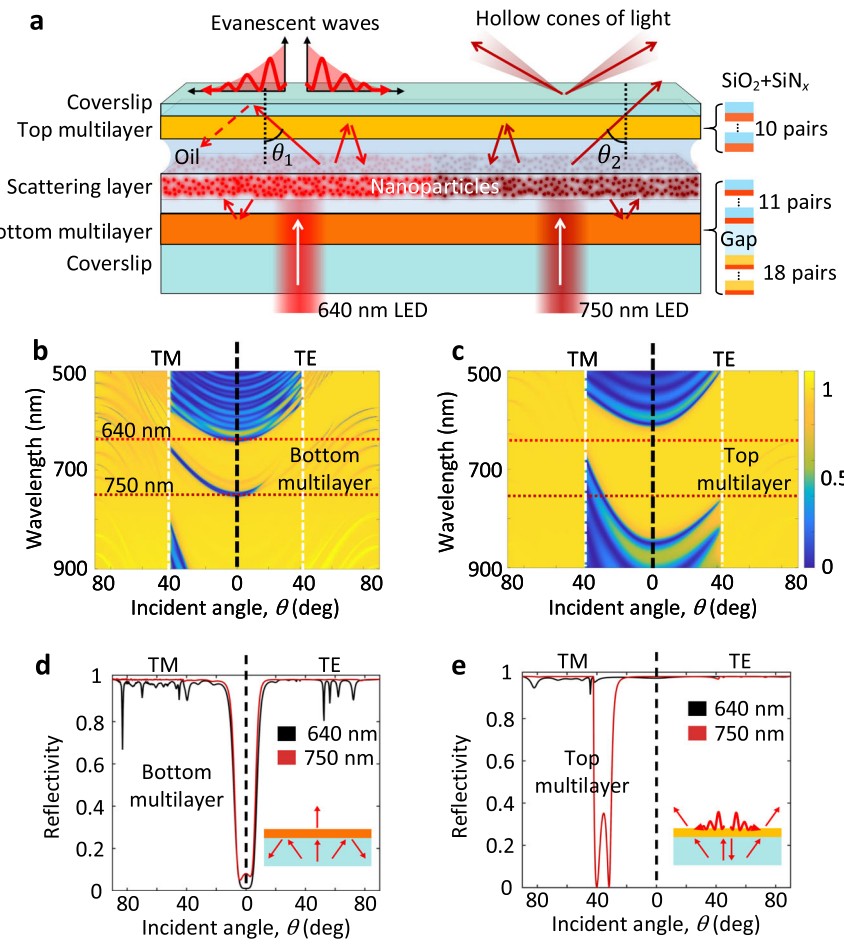

**Fig. 1 Schematic of the planar photonic chip and its photonic band gaps. a** The photonic chips composed of three parts, the bottom (29 pairs of $SiO_2 + SiN_x$ in total), top multilayer (10 pairs of $SiO_2 + SiN_x$ in total), and the scattering layer (doped with $TiO_2$ nanoparticles). Under normal incidence of 640 or 750 nm wavelength light, evanescent waves or hollow cones of light can be generated at the top surface, respectively. **b, c** Calculated PBGs of the bottom and top multilayers. The color scale encodes the theoretical reflectivity of the bottom and top multilayer. The horizontal red-dashed lines represent the positions of the incident wavelengths, 640 and 750 nm. The vertical white-dashed lines represent the position of the TIR angle at glass/air interface (corresponding to NA = 1). The left parts of the **b**, **c** are of the TM-polarized incident beam and the right parts TE-polarized one, separated by the vertical black-dashed lines. **d, e** Calculated angular-dependent reflectivity of the bottom and top multilayers. The incident wavelengths are set as 640 and 750 nm. The incident polarization is of either TM (left part) or TE (right part). The insets on **d, e** simply presents the designed roles of the bottom and top multilayers. TM transverse magnetic, TE transverse electric.

Through properly design of the PBGs, the desired angular transmission of the photonic chip (bottom and top multilayer) can be realized, which is a benefit for high-contrast imaging.

**Fabrication and characterization of the planar photonic chip**. The multilayers were fabricated via plasma-enhanced chemical vapor deposition (PECVD) and characterized with a scanning electron microscope (SEM; Fig. 2a, b). The manufacturing procedures are described in Fig. S3 and section methods. The color scale encoded reflectivity of the bottom and top dielectric multilayer were measured with a reflection back focal plane (R-BFP) imaging set-up[22,23] (Figs. S4–S6) respectively, as shown in Fig. 2c, d, which are consistent with numerical calculations (Fig. 1b, c). The experimental reflection curves at the two selected wavelengths 640 and 750 nm are taken out from Fig. 2c, d, as shown in Fig. 2e, f. On Fig. 2e, the reflection dips appear at the normal incidence (incident angle near 0°, both TM and TE polarization) for the bottom multilayer, which are consistent with dips shown in the simulated curves (Fig. 1d). On Fig. 2f, there are dips appear in the case of TM-polarized light at 750 nm incident wavelength, and the position of dips is consistent with that shown in Fig. 1e (simulated curves). The slightly difference

between the experimental and simulated curves is because the reflectivity for the TE-polarized light gradually decreases in the case of large incident angle, which can be attributed to the depolarization of the reflected light out of the high NA objective used in the BFP imaging experiment (Fig. S4a). Both the simulated and experimental reflectivity curves verify the desired roles of two multilayers.

In the imaging experiments, two low-coherent light-emitting diodes (LED) were used to generate the scattered light from the non-fluorescent $TiO_2$ nanoparticles, which will work as the illumination source for the darkfield image at 750 nm wavelength or TIR images at 640 nm wavelength. Compared to the use of a laser light source for label-free imaging, the LED light will greatly reduce the speckles or interference noise on the optical images, which can further reduce the complexity of microscopes based on our planar photonic chips. A renewed interest in transmitted DM has arisen due to its advantage when used in combination with fluorescence microscopy. Compared to fluorescence emission from quantum dots (QDs) for the illumination source in luminescent-surface-based darkfield imaging[14], the scattered light originating from the high-intensity LED light can excite fluorophores more efficiently than the fluorescence-light from QDs. The LED light source will not

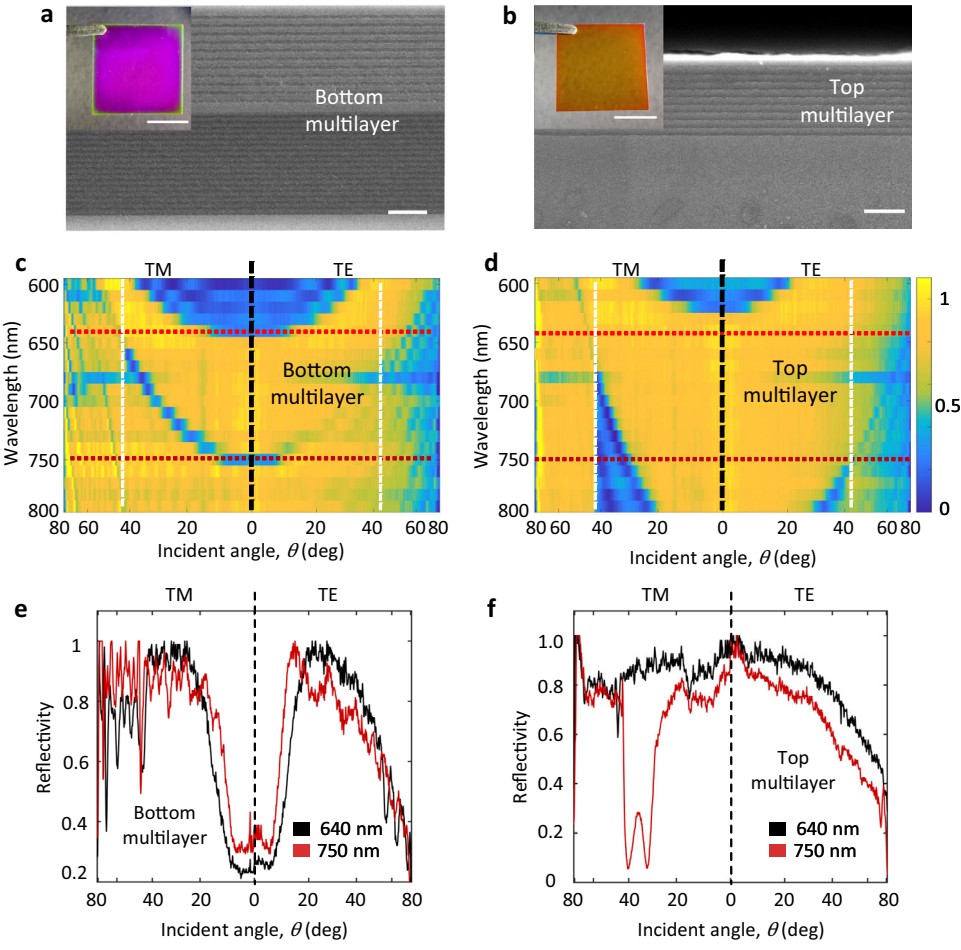

**Fig. 2 The fabricated bottom and top multilayers and their BFP imaging characterizations. a, b** Cross-sectional SEM view of bottom and top multilayers. Scale bar, 1 μm. The insets on **a, b** are the photos of the multilayers fabricated on a coverslip, respectively. Scale bar, 1 cm. **c, d** Measured PBGs of the bottom and top multilayer with the reflective BFP imaging set-up. The color scale encodes the experimental reflectivity of the bottom and top multilayer. The left parts are of the TM-polarized incident beam and the right parts TE-polarized one, separated by the vertical black-dashed lines. The horizontal red-dashed lines represent the positions of the incident wavelengths, 640 and 750 nm. The vertical white-dashed lines represent the position of the TIR angle at glass/air interface, corresponding to the NA = 1. **e, f** The experimental reflection curves for the bottom and top multilayer at the two selected wavelengths (640 and 750 nm). TM transverse magnetic, TE transverse electric.

encounter the problem of photobleaching or blinking that is typical for QDs and other fluorophores, then the proposed C-DFM and C-TIRM will be more stable.

The transmission BFP images of whole photonic chip were measured with the experimental set-up (Fig. 3a), which presents the transmission directions of the scattered light from the TiO₂ nanoparticles. Figure 3 demonstrates three predicted phenomena. First, each point on the BFP image represents one emitting angle (both polar and azimuthal angle) of the scattered light, so the intensity distribution on the BFP images can represent the propagating directions of the scattered light passing through the top multilayer[23]. Second, comparisons between the top and bottom BPF images on Fig. 3d, f, show that the bottom multilayer reflects the light back to the top multilayer, otherwise this light would have been lost. Thus, it results in much more intense light exiting from the top multilayer at the designed polar angles, and more intense evanescent waves, which benefit both darkfield and TIR imaging. Third, the intensity of the scattered light within NA < 0.70 and that with NA from 0.70 to 1.49 were derived from Fig. 3d, f, as shown in Fig. 3e, g. It shows that this chip can efficiently prevent the direct transmission at lower NA (corresponding to small polar angle), and

most of transmission energy (about 98.5%) was localized inside the large NA regions (0.7–1.49), which is favorable for the contrast of DFM or TIRM images.

**The planar photonic chip working as a high-contrast imaging device.** In the following experiments, a polymer nanowire and polymer microwire were used as the specimens, although they can certainly be replaced with real cells. The nanowire is approximately 70 nm in diameter, and the microwire is approximately 3–4 μm in diameter (Fig. S7b). The single polymeric nanowire was located on a polymeric microwire that are both placed on a coverslip. A regular air objective (×40, NA 0.60) was used for the standard brightfield imaging under the normal illumination of LED light at 640 and 750 nm wavelengths, and the brightfield images are shown in Fig. 4b, f. Second, when the photonic chip was attached below this coverslip with index-matched oil, the polymer wires on the C-DFM and C-TIRM images becomes more distinguished due to the imaging contrast (CR) enhancement, as indicated in Fig. 4e, i. When the illumination wavelength was 750 nm, the scattered light from the chip will be out of the NA of the objective. The images (Fig. 4c, d) showed typical darkfield

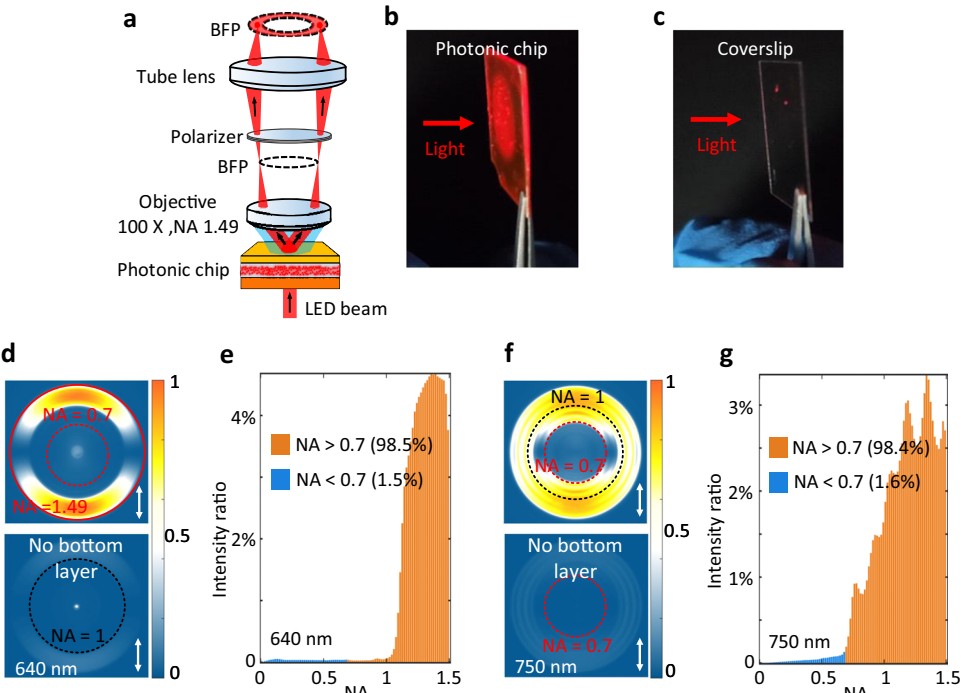

**Fig. 3 Measuring the transmissive directions of the scattered light from the photonic chip. a** Schematic of the experimental set-up for the transmissive BFP imaging. **b**, **c** The photos of the photonic chip and a coverslip when a light beam passes through, which show that light field can be generated at the surface of the photonic chip. **d**, **f** Transmission BFP images of the photonic chip under normal incidence with a light-emitting diode (LED) beam, the incident wavelengths are selected as 640 and 750 nm. The color scale encodes the experimental transmission power (normalized) of the bottom and top multilayer. The bottom panels of **d**, **f** are the case where the bottom multilayer of the photonic chip was removed, to demonstrate the role of bottom multilayer. Comparisons between top and bottom panels of **d**, **f** shows that the bottom multilayer can enhance the intensity of the transmissive light from the photonic chip. The red-dashed circle represents the position with NA = 0.7 (for a regular air objective used in the darkfield and TIR imaging experiments) and the black one NA = 1 (corresponding to the TIR angle), the oil-immersed objective's numerical aperture (NA) is marked with a red-solid circle (NA = 1.49). The orientation of the polarizer is marked with a solid arrow on **d**, **f**. **e**, **g** Quantitatively demonstrating the angular distribution of the transmissive light from the photonic chip. The NA of the horizontal axis is corresponding to transmission angle θ (NA = $n \times \sin(\theta)$, $n$ is the refractive of the oil). The intensity of the transmissive light within low NA (<0.7) is about 1.5% (or 1.6%), meaning that the direct transmission of the scattered light from the photonic chip is very weak.

characters, a bright image of the specimens superimposed onto a dark background. The contrast (CR) of the darkfield image, calculated as the difference between the maximum and minimum image intensity values divided by their sum (Fig. 4e, f), was significantly improved when comparing with that of the brightfield image (Fig. 4e). The polymer wires on Fig. 4d are brighter than those on Fig. 4c, verifying that the bottom multilayer can recycle the scattered light and enhance the intensity of the scattered light that transmit through the top multilayer at the designed polar angles, which are consistent with the transmission BFP images (Fig. 3d, f).

Except for the differences in contrasts, the darkfield images of the nanowire appear discontinuous when it crosses the microwire. This phenomenon can be explained as follows. When the illumination wavelength is 750 nm, the specimen will be illuminated not only by the transmitting light at oblique polar angles (0.70 < NA < 1.0) and at all azimuths, but also evanescent waves (NA > 1.0) on the coverslip-air interface induced by the TIR, as shown in Fig. 3d. In this crossing area, the nanowire is on the microwire and is far away from the coverslip. Due to the longitudinal decay of evanescent waves[24], this cross-section will not be imaged. On the contrary, this longitudinal air gap cannot be discovered from brightfield images (Fig. 4b, f). This phenomenon will be more obvious, as that the nanowire is nearly invisible in the crossing area, when the wavelength was changed to 640 nm (Fig. 4g, h). In this case, the specimens are

illuminated only by pure evanescent waves, and the BFM was transformed to a C-TIRM with the aid of this photonic chip as an add-on.

When the polymer wires were replaced with polymer particles, the images enhancement induced by the photonic chip is also obvious (Fig. S8). The edge of the microparticles is shaper on the TIR image (Fig. S7c) than that on the darkfield image (Fig. S7b), because that the TIR imaging uses higher-spatial-frequency component of the illumination source and can provide higher spatial resolution. The phenomena on Fig. 4f–i verify that, using this photonic chip below the substrate, the TIRM images can be captured by a standard BFM, which can focus on the targets within the evanescent field (100–200 nm), rather than those contained in the entire sample. It should be noted that the photonic chip still works for darkfield and TIR imaging when the specimens are immersed in water or other liquid solution. The refractive index contrast between the polymer wires and water is smaller than that between the polymer wires and air. As shown in Figs. S9 and S10 of Supplementary Information, the polymer wires immersed in water solution, and live biological cells (CT26, a murine colorectal carcinoma cell line which is from a BALB/c mouse) cultured in Dulbecco's Modified Eagle Medium (DMEM) with 10% serum, are placed on the photonic chip. Darkfield imaging and TIR imaging of these specimens are realized by the conventional BFM with the aid of this photonic chip. The ability for darkfield and TIR imaging of the specimens in water using the

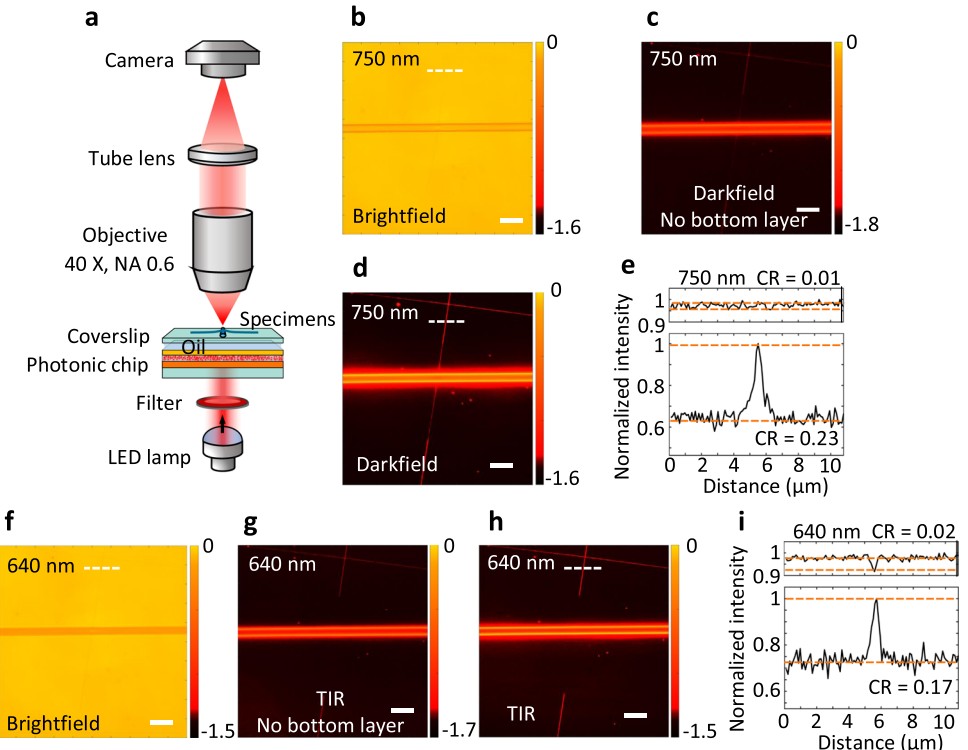

**Fig. 4 Darkfield and total-internal-reflection imaging enabled by the photonic chip. a** Schematic of the brightfield microscopy with the photonic chip. The specimen to be imaged is a polymer nanowire placed on a polymer microwire. **b, f** The brightfield images of the specimens, when they were placed on a bare coverslip. **c, d** C-DFM images, **g, h** C-TIRM images. For **c, g**, the bottom multilayer of the photonic chip was removed, to demonstrate the role of the bottom multilayer in the C-DFM and C-TIRM. **e** Intensity profiles extracted along the white dashed lines on **b, d. i** Intensity profiles extracted from **f, h**. The red lines indicate the levels used to determine the image contrasts (CR). From **b** to **e**, the incident wavelength is 750 nm, and from **f** to **i**, the wavelength is 640 nm. Scale bars, 20 μm. TIR total internal reflection.

proposed photonic chip will have high potential impact on the microscopy community which is more likely to look for samples immersed into water, or with high magnification aperture which requires water or oil optical coupling.

The light coupling efficiency of the photonic chip was measured with the experimental set-up as shown in Fig. S11. The light beam from the LED with divergent angles at 3.7° (for 640 nm wavelength) and 10.0° (for 750 nm wavelength) was normal incident onto two control substrates, one is the bare glass substrate, and the other is the designed photonic chip as shown in Fig. 1a. An upright objective (NA 1.49, ×100) was used to collect the transmitted light beam whose power was measured with a power meter. For the substrate made of bare glass substrate, the power of the transmitted light is 30.06 μW (at 640 nm wavelength) or 9.04 μW (at 750 nm wavelength). For the substrate made of the photonic chip, the power of the transmitted light is 8.80 μW (at 640 nm wavelength) or 4.5 μW (at 750 nm wavelength), then the coupling efficiency can be calculated as 29% (at 640 nm wavelength) or 50% (at 750 nm wavelength). The NA of the objective used to collect the transmitting light is only 1.49, and the transmitted light out of this light-cone cannot be collected, but also contributes to the darkfield and TIR imaging, so the real coupling efficiency is a little larger than the measured result.

For the conventional DM where an Abbe condenser is used, an opaque spider-style light stop is inserted below the Abbe condenser, the central light rays are blocked, allowing only peripheral light rays to pass through the lenses to form an inverted oblique hollow cone of light. This form of illumination is wasteful of light and thus demands a high intensity illumination source, meaning that the coupling efficiency of the conventional

DM is much lower. For the TIR imaging where a high refractive index prism is used to generate the evanescent waves, the coupling efficiency can be as high as 100% if the light path is precisely aligned. However, the precise alignment needs additional components which will add cost and complexity to the microscopy. The prism is also bulky and not suitable for optical integration or compact use. On the contrary, the proposed photonic chip is planar and compact, and the normal incidence does not need specialized component. The coupling efficiency can be further enhanced with optimized parameters of the photonic chip, such as that the narrower angular divergence of the dip (shown in Fig. 1d) can induce high coupling efficiency in the case of normal incidence. Due to the planar shape of the photonic chip, in future, the LED light source can be integrated below this chip, it will be more compact for integration and the coupling efficiency can be further enhanced.

**Excitation of large-scale surface plasmons (SPs) with the planar photonic chip.** Furthermore, it can be anticipated that this photonic chip has the potentials to excite other kinds of surface waves under normal incidence, such as SPs and Bloch surface waves (BSWs) that are more sensitive to environmental changes than the evanescent waves used in TIR fluorescence microscopy and can provide enhanced local fields[25]. Typically, either a bulk prism, or a nanofabricated structure, or an oil-immersed objective is required for the excitation of SP or BSWs, which results in either a bulky and complicated system or limited excitation areas[26]. However, these limitations can be removed by using the photonic chip. To verify this point, the top multilayer was replaced with a coverslip coated with a 55-nm-thick Ag film (Fig. 5a). A regular air objective (×60, NA 0.70) was used to

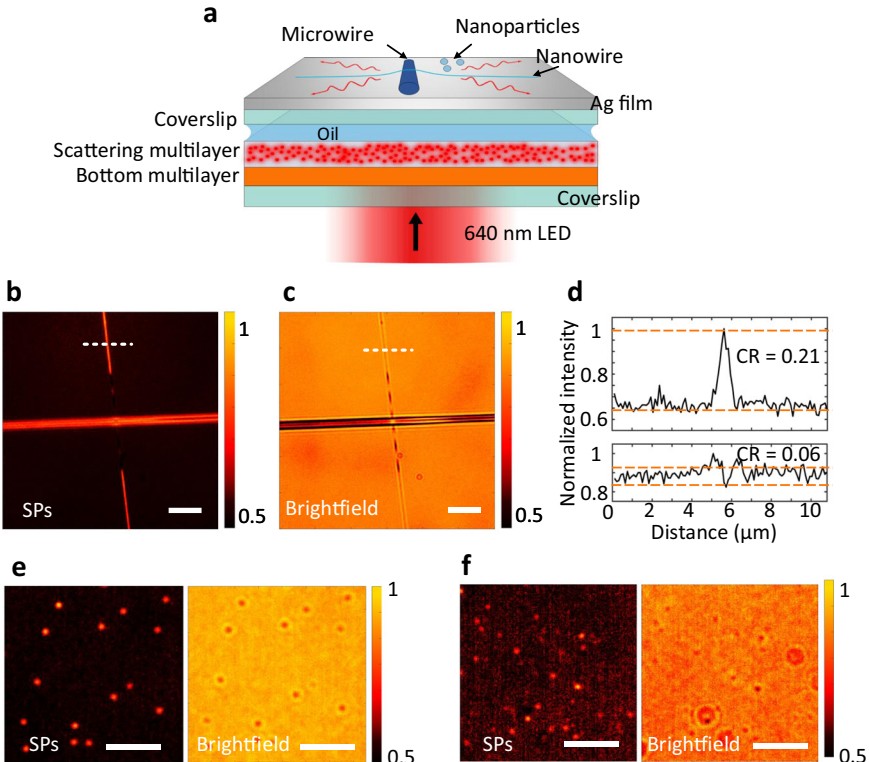

**Fig. 5 Excitations of SPs with the photonic chip for surface imaging. a** Schematic of the modified photonic chip where the top multilayer was replaced with a thin silver film to demonstrate the excitation of SPs under normal incidence and the ability for surface imaging. The specimens to be imaged are the single polymer nanowire placed on a microwire, and polymer nanoparticles of different diameters. The modified photonic chip was also used in the standard brightfield microscopy shown in Fig. 4a with a regular air objective (×60, NA = 0.7). **b** The image of the polymer wires under the illuminations of the excited surface plasmons (SPs). **c** Corresponding brightfield image of the polymer wires. **d** Intensity profiles extracted along the white dashed lines on **b**, **c**. The dashed red lines indicate the levels used to determine the image contrast (CR). **e**, **f** The surface images and brightfield images of the polymer nanoparticles with diameter at 200 and 50 nm, respectively. Scale bars, 10 μm.

captured the images with the same experimental set-up as shown in Fig. 4a. The images (Fig. 5b) show a typical character of surface-sensitive patterns with enhanced imaging contrast (Fig. 5d), where the nanowire appears discontinuous in the cross-region with the microwire. This phenomena verifies the excitation of the SPs. The advantage of images illuminated by SPs over the brightfield images will be obvious when the diameter of the polymer nanoparticles changes from 200 to 50 nm (from Fig. 5e, f). When the diameter of the polymer nanoparticles is about 50 nm, these nanoparticles are nearly invisible on the brightfield image (Fig. 5f), but are visible on the image illuminated by the SPs (Fig. 5e). The excitation of the SPs with the normal incident light will make the optical path easily aligned and simplifies the configuration. The use of a planar chip means that the excitation areas of the SPs can be much larger than that excited with in-plane nanostructures (such as the inscribed gratings[27]), which is favorable for high throughput imaging, sensing and tracking of the specimens. Different from the dielectric film used in Fig. 4, the silver film is conductive, then the proposed experimental configuration has the potential to work as an electrochemical microscopy for imaging local electrochemical current, studying heterogeneous surface reactions and for analyzing trace chemicals[28].

## Discussion

In summary, through placing a photonic chip below the specimens, a standard BFM can be easily transformed to both a C-DFM and C-TIRM without modifying the configuration or the specimens. The principle lies on that, under normal incidence, both the inverted hollow cones of light and various evanescent waves can be generated with this photonic chip due to its tailored angular transmission. The roles of this photonic chip working as a high-contrast imaging device, are confirmed by both theoretical and experimental results. Thus, it demonstrates the potential of the proposed imaging device for a novel type of versatile and compact microscope. The working wavelengths of the photonic chip can be tuned through changing thickness and refractive index of the dielectric layers; thus, the devices enable multi-spectral darkfield and TIR imaging using simple brightfield microscopes[14]. If the specimens are fluorescent, fluorescence imaging also can be realized, similar as the TIRF[29]. The set-up can perform simultaneous C-DFM/C-TIRM and fluorescence microscopy of the same specimen, and thus make it possible to combine the strengths of both labeled and label-free detection and fluorescent imaging technologies in one integrated set-up[30–32]. When combining with stochastic optical reconstruction technique, the C-TIRM will have the potential to be a chip-based wide field-of-view nanoscopy[2]. It has proven affordable and easy for users to launch as an add-on to a regular brightfield microscope, thus, it will make full use of the BFM that can be found in many academic and industrial labs.

When comparing with bulk Abbe condensers used in a conventional DFM, and prism or oil-immersed objective used in conventional TIR imaging, this imaging planar chip device is more compact, low cost and easy aligned. It can be fabricated on an extremely large substrate with standard deposition and spin-coating method, without any top–down nanofabrication

procedures, thus open new avenues towards the design of a fully integrated on-chip microscopy. Different from the condenser or objective that has limited illumination area (of micrometer scale), this imaging device enables extremely large illumination area up to centimeter-scale or even with microwell plate readers with dimensions $85 \times 128$ nm. In the future, the combination of photonic chip with micro lens arrays for light collection have the potential for extraordinarily high throughput, with illumination and collection done in parallel over large areas, completely removing the dependency on a bulky objective lens[3].

## Methods

**Fabrication of the planar phonic chips working as the compact imaging devices**. The top and bottom dielectric multilayers were fabricated via PECVD (Oxford System 100) of $SiO_2$ and $SiN_x$ on a coverslip (0.17 mm thickness) at a vacuum 0.1 mTorr and temperature of 300 °C. Before the PECVD of a dielectric multilayer, the coverslip was cleaned with piranha solution and then with nanopure deionized water and dried with an $N_2$ stream. The process of PECVD depends on the chemical reaction of $SiH_4$ with $N_2O$ and $NH_3$ at high temperature. The refractive index of $SiN_x$ can be adjusted from 1.9 to 2.4 by changing the ratio of $SiH_4$ to $NH_3$. The $SiO_2$ layer is of the low (L) refractive index dielectric and the $SiN_4$ layer is the high (H) one. Thickness of each layer was presented in details in Fig. S1. There are 18 pairs of $SiN_x$ (2) + $SiN_x$ (3) and 11 pairs of $SiO_2$ + $SiN_x$ (1) in total for the bottom multilayer. There are 10 pairs of $SiO_2$ + $SiN_x$ (1) in total for the top multilayer. The top and bottom multilayer were fabricated on two independent coverslips for the BFP imaging experiments to measure the PBG of the multilayer (Figs. 2 and 3).

The spacer layer between the top and bottom multilayer is made of Intermediate Coating IC1-200, which is a polysiloxane-based spin-on dielectric material. The IC1-200 solution doped with $TiO_2$ nanoparticles (diameter at about 60 nm) was then spin-coated on to the bottom multilayer, which will work as the scattering layer to generate scattered light of various propagating directions. Thickness of the spacer layer is about 2 μm and its refractive index is about 1.41. The SEM image of the scattering nanoparticle $TiO_2$ was shown in Fig. S7a.

For the darkfield and TIR imaging experiments, the three parts of the planar photonic chip will be assembled together with the refractive index matched oil and then form the unit substrate (Fig. 4a). The silver film was deposited on the bare coverslip with the thermal evaporation method, whose thickness is about 55 nm. For the surface imaging with SPs (Fig. 5), this coverslip coated with Ag film was attached onto the spacer (scattering) layer with refractive index matched oil.

It should be noted, this photonic chip still can be used to realize the darkfield and TIR imaging if the bottom multilayer was removed, as shown Fig. 4c, g. However, the obtained darkfield and TIR images will be much weaker than those (Fig. 4d, h) obtained with the whole photonic chip (top multilayer + scattering layer + bottom multilayer). Or in other words, the bottom multilayer can recycle scattered light into propagation angle ranges that are transmitted by the top multilayer, and then enhance the light intensity on or out of the top multilayer, which was verified by the comparisons between the top and bottom images on Fig. 3d, f.

**Preparation of the polymer wires and nanoparticles and live biological cells as the specimens to be imaged**. The typical procedure for the fabrication of electrospun microfibers and nanofibers is given below. A 2 ml formic solution (solvent for the Nylon) containing 1.6 g Nylon 6 was ejected at a continuous rate using a syringe pump through a stainless-steel needle. A voltage of 10 kV was applied to the needle with a high voltage power supply and a feed rate of 0.2 mm per minute was maintained with a syringe pump. A collector (the glass substrate with the fabricated dielectric multilayer) was placed at 10 cm from the needle tip to collect the polymer nanowires. By replacing the solution in the syringe pump into a tetrahydrofuran solution containing 0.25 g hydrogel, the hydrogel microwire can be produced with the same procedure. The SEM images of the polymer wires are shown in Fig. S7.

The polystyrene nanoparticles (Figs. 5 and S8) were purchased from Thermo Fisher Scientific (USA). The certified mean diameters of the nanoparticles supplied were about 20 nm, 50 nm, 100 nm, 200 nm and 2 μm. The SEM images of these nanoparticles are shown in Fig. S7. The nanoparticles dispersed in water are spin coated on the substrate. After dried by hot plate, the nanoparticles are fixed on the surface of the substrate. The live biological cell is CT26, a murine colorectal carcinoma cell line which is from a BALB/c mouse, and the cells for imaging experiments are cultured in DMEM with 10% serum.

In the darkfield and TIR imaging (Figs. 4 and S8–S10), the polymer wires, polystyrene nanoparticles and live biological cells were placed on a clean coverslip, which was then attached to the top surface of the photonic chip (Figs. 1a and 4a) with refractive index matched oil between them. However, the specimens (wires and particles) can also be put on the top surface of the photonic chip directly for darkfield and TIR imaging, then this planar photonic chip both holds and illuminates the specimen. In the brightfield imaging used for comparisons, this

photonic chip was removed. For the surface imaging with SPs (Fig. 5), the wires and nanoparticles were placed on the silver film directly.

**Optical characterization set-up**. All optical measurements were performed on modified optical microscope (Nikon Ti2-U). For the reflection BFP imaging set-up (Fig. S4), an oil immersion objective (CFI Apochromat TIRF ×100, NA = 1.49, WD = 0.12 mm) from Nikon, Japan was used to fully measure the reflecting angular distribution, which corresponds to the polar angle ranging from −80° to 80° in the oil medium. To minimize the interference fringes on the BFP images, we used a noncoherent light (tungsten bromine lamp combined with a serials of band-pass filters) as the illumination source. The center wavelengths of the band pass filter ranges from 600 to 790 nm (20 filters in total), with a full width at half maximum (FWHM) of 10 ± 2 nm. The Neo sCMOS detector for recording the BFP images was from Andor Oxford Instruments (UK). By properly tuning the distance between the tube lens and the detector, BFP image of the objective can be recorded. At each incident wavelength, one BFP image can be obtained. From all these BFP images (Figs. S4–S6), PBGs of the top and bottom multilayers can be derived.

For the transmitted BFP images (Fig. 3a), the illumination source was changed to LED with center wavelength at 640 and 750 nm. Two bandpass filters (FWHM of 10 ± 2 nm, center wavelengths of 640 and 750 nm, Thorlabs Inc.) were used to select the required emission wavelengths from the LED sources. Under normal incidence, the transmitting angular distribution of the scattered light escaping out of the photonic chip was measured with the same objective with NA at 1.49. The detector and tube lens are the same as those used in reflection BFP imaging set-up. It should be noted, in the BFP imaging of the photonic chip and independent multilayer, no specimens (polymer wires and polystyrene particles) were put on the chip.

In the darkfield and TIR imaging of the specimens, two regular air objectives (CFI Super Plan Fluor ×60, NA = 0.70, WD = 2.61–1.79 mm and ×40, NA = 0.6, WD = 3.6–2.8 mm) were used for the brightfield, darkfield, and TIR imaging experiments. The distance between the tube lens and the detector was changed so that the front focal plane of the objective can be imaged. The LED with bandpass filter was used as the illumination source.

**Reporting summary**. Further information on research design is available in the Nature Research Reporting Summary linked to this article.

## Data availability
The data that support the plots within this paper and other finding of this study are available from the corresponding author upon reasonable request. Source data for Figs. 1–5 are available at https://doi.org/10.6084/m9.figshare.16842913.

## Code availability
The codes that support the findings of this study are available from the corresponding author upon reasonable request.

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

## Acknowledgements

This work was supported by the National Nature Science Foundation of China (grant nos. 12134013, 11774330), the Anhui Initiative in Quantum Information Technologies (grant no. AHY090000), Advanced Laser Technology Laboratory of Anhui Province (grant no. 20192301), Hefei Municipal Natural Science Foundation (grant no. 2021007), and Key Research & Development program of Anhui Province (202104a05020010). J.R.L. thanks the National Institute of General Medical Sciences for support under grant nos. R01 GM125976 and R21 GM129561 and the National Institutes of Health for support under grant nos. S10OD19975 and S10RR026370. The work was partially carried out at the University of Science and Technology of China's Center for Micro and Nanoscale Research and Fabrication. D.Z. is supported by a USTC Tang Scholarship.

## Author contributions

D.Z. initiated the work, supervised the project, and wrote the manuscript. Y.K., Z.F., and D.Z. carried out the optical experiments and fabricated the samples; Y.K. and J.C. carried out the theoretical calculations. G.Z. contributed to the fabrications of polymer wires. J.R.L. assisted in clarifying and revising the manuscript. All authors discussed the results and commented on the manuscript.

## Competing interests

The authors declare no competing interests.
