## [Peer Review File · Nature Communications]

REVIEWER COMMENTS

Reviewer #1 (Remarks to the Author):

Authors present the design, fabrication, and experimental demonstration and evaluation of multiple multilayered coverslip for dark field and TIR microscopy. The overall work is highly interesting and I did not find any (obvious) mistake preventing the trust in the results. It is clear that authors have performed an excellent work and obtained very neat proof of their concept.

The work must be compared to the study published in 2020 in Nature Photonics (cited as ref 14) by Chozet et al. In this novel work, I see two major differences: 1) the use of a bottom multilayer to reflect back to the surface the scattered light from the titania nanoparticles, and 2) the use of titania nanoparticles. These two points are clear improvements in the microscope chip that authors propose and pave the way to tremendous research in the field of microscopy.

All claims are supported by results and interpretations and conclusions are sound (except details, see below).

All details are provided so that the experiments can be conducted in another laboratory.

Comments:

1) There are numerous typos and other text issues and I recommend a proof reading by a native English speaker:

a) PBG appears line 81 without being defined earlier.

b) TiO₂ appears several times with 2 as a subscript.

c) I suggest to use the words "scattered light" instead of "scattering light".

d) When several figures are cited, I suggest to use a "s" at the end of the word "Figure".

e) NA, N.A., or N. A.: Author must choose, but must never use "N.A". This must be corrected.

e) Mathematical signs (>, <, =, for instance) are always preceded and followed by a space. This must be corrected.

f) Parameters are usually in italic.

g) Line 145: "It clearly shows..." replace by "It shows".

h) Line 163: "... is brighter..." replace by "... are brighter...".

i) Line 222: "has the potential" replace by "have the potential".

2) Line 119: Authors state that the experimental results are "nearly consistent with numerical calculations". I suggest to be less modest with such a clear excellent match. However, this sentence pushed me to look carefully at Figures 2c and 2d. I noted that there is a big issue with the label of the y axes (80 60 instead of 60). I hope this is simply a mistake... This must be corrected thoroughly.

3) At many places, authors are using the word "amplify" to tell that more light is reaching the detector (camera of the microscope setup). This word is misleading because there is no gain medium in their chip.

4) Lines 140-141: "... the intensity of the scattered light that reaches the top multilayer." This sentence is not really good. I would rephrase it to explain that the bottom multilayer reflects back to the top multilayer light that would have been lost otherwise.

5) Lines 156-157: "...are much more defined." does not sound scientific. A quantitative value would be appreciated.

6) A simplified version of the very long sentence on lines 160 to 162 would be: "The dark field

image contrast was improved compared to that of the brightfield image". I suggest to use clearer, although heavier, wordings like " the contrast of the dark field image...".

7) Line 184-185: "..., using this high compatible photonic chip below the substrate, ..." sounds like a free statement.

8) Lines 196-198: "typical character of surface sensitive patterns" does it refer to the discussion in the paragraph between lines 167 and 178.

9) Line 205: "such as the inscribed gratings" requires a reference.

10) Lines 222-223: "The imaging device can be washed, sterilized, and used multiple times". This must be proven.

Reviewer #2 (Remarks to the Author):

The paper presents an innovative concept for improving contrast in a microscope. However, I miss a comparison with existing systems, such as phase contrast, differential interference contrast, etc. I would expect light to be lost through the scattering layer. This still needs to be analyzed.

Also, it is not clear to me if the paper is mainly about the photonic chip, or more about a dedicated device for microscopy. The combination of multilayer and surface plasmons has been shown in other work.

I think the work has innovative parts that should be published. However, for publication in Nature Communication, the work should focus more on the main innovation. So please emphasise the main innovation. Also a discussion about the coupling efficiency compared to other concepts should be added.

Reviewer #3 (Remarks to the Author):

The manuscript by Yan Kuai and coworkers describes a compact photonic system to replace the condenser lens issued in dark-field and total internal reflection microscopy (TIRM). Their system is based on a stack of three different planar photonic chips, which are well motivated and characterized in the paper. The first layer system works as an angular filter to collimate the light from the LED source. The second layer is based on dielectric nanoparticles to scatter the light in all directions and the third last layer is again an angular filter. Depending on the illumination wavelength, the light finally transmitted to the sample occurs at high angles (for dark-field microscopy at 750 nm) or corresponds to evanescent waves (for TIRM at 640 nm). I believe this approach is quite original and I am unaware of anything similar.

Altogether, while I find the concept original and interesting, I somehow miss a valuable groundbreaking application here. The authors essentially show images of dielectric nanowires and 200 and 50 nm polystyrene nanoparticles in air, thereby taking advantage of the high refractive index contrast between the polystyrene and air. That makes me seriously question the potential impact of this work on the microscopy community which is more likely to look for samples immersed into water.

The paper is nicely written but honestly I find this is currently more the type of work found in ACS Photonics and related journals. I would like to recommend publication in Nature Communications, but then I would require the authors to show a more compelling demonstration of the merit of their system on a more challenging sample for imaging. For instance, working on a biological cell sample in water medium and using TIRM eventually combined to fluorescence to image the basal cell membrane only. This is what TIRFM is typically used for. An even more potentially impressive application would go to UV illumination of fluorescence, where the choice of optical condenser elements is much more restricted (but UV-LED with quite high power are available). However, this implies changing completely the wavelength range used by the authors so far, and I would understand if the authors prefer to stay in the red spectral they are familiar with.

Other comments:

- Many sub-figures lack a description of the quantity plotted (Fig. 1b&c, 2c&d, 3d&f...). Sometimes it is the reflectivity, sometimes the transmission. This should be clarified every time.

- I miss to see a description of the numerical simulation methods and approach

- I would like to see a comparison between experiments (Fig. 2c,d) and simulations (Fig. 1b,c) for the two selected wavelengths 640 & 750 nm. Why not remove Fig. 1b,c to the SI and add the experimental characterization data at the bottom of Fig. 1 in the same fashion as Fig. 1d,e? I believe this would convince the reader more easily.

- line 459, "Light filed...", do the authors mean "light field"?

- Fig. 3e&g, I do not understand the vertical scaling and why it is only a few percent. A few percent of what reference? That leads to the question: how much light is lost by this condenser? What is the available power transmitted to the sample?

- I am not a great fan of Figure 5. I understand the idea to show this system is also able to generate surface plasmon waves on a thin silver interface, but I find that (1) this is quite obvious and expected and (2) this is introducing a different system into the story, so that the reader then asks the question which device is the best and what is the interest for the other one then?

RESPONSES TO REVIEWERS:

Reviewer #1 (Remarks to the Author):

Authors present the design, fabrication, and experimental demonstration and evaluation of multiple multilayered coverslip for dark field and TIR microscopy.

The overall work is highly interesting and I did not find any (obvious) mistake preventing the trust in the results.

It is clear that authors have performed an excellent work and obtained very neat proof of their concept.

The work must be compared to the study published in 2020 in Nature Photonics (cited as ref 14) by Chozet et al. In this novel work, I see to major differences: 1) the use of a bottom multilayer to reflect back to the surface the scattered light from the titania nanoparticles, and 2) the use of titania nanoparticles. These two points are clear improvement in the microscope chip that authors propose and pave the way to tremendous research in the field of microscopy.

All claims are supported by results and interpretations and conclusions are sounded (except details, see below).

All details are provided so that the experiments can be conducted in another laboratory.

Comments:

1) There are numerous typos and other text issues and I recommend a proof reading by native English speaker:

a) PBG appears line 81 without being defines earlier.

b) TiO_2 appear several times with 2 as a subscript.

c) I suggest to use the words "scattered light" instead of "scattering light".

d) When several figures are cited, I suggest to use a "s" at the end of the word "Figure".

e) NA, N.A., or N. A.: Author must chose, but must never use "N.A". This must be corrected.

e) Mathematical signs (>, <, =, for instance) are always preceded and followed by a space. This must be corrected.

f) Parameters are usually in italic.

g) Line 145: "It is clearly show..." replace by "It shows".

h) Line 163: "... is brighter..." replace by "... are brighter...".

i) Line 222: "has the potential" replace by "have the potential".

Response: Thank you very much for your suggestions. We really appreciate that the reviewer has read our manuscript so careful, and provided us so many valuable reminders and suggestions. I have revised the main text, the supplementary materials and the figures accordingly. All the revisions in the main text are marked in blue. Prof Lakowicz, who is a native English speaker has conducted a proof reading of the revised manuscript.

中國科學技術大學

UNIVERSITY OF SCIENCE & TECHNOLOGY OF CHINA

2) Line 119: Authors state that the experimental results are "nearly consistent with numerical calculations". I suggest to be less modest with a so clear excellent match. However, this sentence pushed me to look carefully at Figures 2c and 2d. I noted that there is a big issue with the label of the y axes (80 60 instead of 60). I hope this is simply a mistake... This must be corrected thoroughly.

Response: Thank you for the suggestion. We have revised this sentence and remove the word "nearly".

The label of horizontal axis on Figures 2c and 2d is the incident angle θ . The vertical axis is the incident wavelength. The color scale encodes the normalized reflectivity of the light reflected from the multilayer.

Please see Figure S4 (for your convenience, we copy Figure S4 in the following). The incident angle θ was calibrated from the known NA of the objective (used for the Back focal plane imaging, NA=1.49, Figure S4a) and radius of the selected point on this BFP image. For example, the incident angle for point A (Figure S4b) can be derived with the following equation.

$$\theta_A = \sin^{-1} \left(\frac{r_A * NA}{R * n_{oil}} \right)$$

It clearly shows that the incident angle (θ_A) is not linearly changed with the radius r_A . More closed to the boundary of the BFP image (larger r), quicker changes of the angle (θ).

Please see the direct illustration of this change in the following BFP images (labeled with angle) that is adopted from our previous publication (Nat Commun 8, 14330 (2017)).

As shown in Figure S4c, the normalized reflectivity (vertical axis) is derived from the image intensity of each point along the dash lines on Figure S4b (marked with TM and TE, meaning the incident polarization), which is then plotted vs. the incident angle (horizontal axis). As a result, the angle on the horizontal axis is not of uniformly-spaced. The distance between 80° and 60° is closer than that between 60° and 40° .

Figures 2c and 2d are formed from the reflectivity curves (shown in Figure S4c) at each incident wavelength. The incident angle on the horizontal axis of Figures 2c and 2d is also not of uniform-spaced, so you can see the label 80° is much closed to the label 60° .

[redacted]

3) At many places, authors are using the word "amplify" to tell that more light is reaching the detector (camera of the microscope setup). This word is misleading because there is no gain medium in their chip.

Response: Thank you very much for your comment. Yes, the word "amplify" is misleading because that there is no gain medium here. I replace the word "amplify" with the word "enhance".

4) Lines 140-141: "... the intensity of the scattered light that reaches the top multilayer." This sentence is not really good. I would rephrase it to explain that the bottom multilayer reflects back to the top multilayer light that would have been lost otherwise.

Response: Thank you very much for your suggestions. I have revised this sentence as following.

Secondly, comparisons between the top and bottom BPF images on Figures 3d and 3f, show that the bottom multilayer reflects the light back to the top multilayer, otherwise this light would have been lost. Thus, it results in much more intense light exiting from the top multilayer at the designed polar angles, and more intense evanescent waves, which benefit both darkfield and TIR imaging.

5) Lines 156-157: "...are much more defined." does not sound scientific. A quantitative value would be appreciated.

Response: Thank you for your suggestion. We have revised this sentence as following.

the polymer wires on the C-DFM and C-TIRM images becomes more distinguished due to the imaging contrast (CR) enhancement, as indicated in Figures 4e and 4i.

6) A simplified version of the very long sentence on lines 160 to 162 would be: "The dark field image contrast was improved compared to that of the brightfield image". I suggest to use clearer, although heavier, wordings like " the contrast of the dark field image...".

Response: Thank you for your suggestion. I have revised this sentence based on your suggestion.

The contrast of the darkfield image, calculated as the difference between the maximum and minimum image intensity values divided by their sum (Figures 4e and 4i), was significantly improved when comparing with that of the brightfield image (Figure 4e).

7) Line 184-185: "..., using this high compatible photonic chip below the substrate, ..." sounds like a free statement.

Response: Thank you for your reminder. Yes, this description sounds like a free statement and is not suitable for your scientific paper. I remove the words "high compatible" from this sentence.

The new sentence is as following.

The phenomena on Figures 4f-4i verify that, using this photonic chip below the substrate, the TIRM images can be captured by a standard BFM,

8) Lines 196-198: "typical character of surface sensitive patterns" does it refer to the discussion in the paragraph between lines 167 and 178.

Response: Thank you for your reminder. Yes, it refers to the discussion in the paragraph between lines 167 and 178. The typical character of surface sensitive patterns means that the images of the nanowire appear discontinuous when it crosses the microwire.

9) Line 205: "such as the inscribed gratings" requires a reference.

Response: Thank you very much for your suggestion. We have added a reference for this description.

27. Zayats AV, Smolyaninov, II. Near-field photonics: surface plasmon polaritons and localized surface plasmons. *Journal of Optics a-Pure and Applied Optics* 5, S16-S50 (2003).

10) Lines 222-223: "The imaging device can be washed, sterilized, and used multiple times". This must be proven.

Response: Thank you very much for your reminder. In our experiment, the photonic chip made of all-dielectric layer can be used for many times. For examples, this chip was used for one-by-one imaging the polymer wires, dielectric particles of various diameters, as shown in Figures 4, S4, and S8. For imaging each target, the chip was washed and

中國科學技術大學

UNIVERSITY OF SCIENCE & TECHNOLOGY OF CHINA

sterilized for the next round of imaging. However, for the photonic chip containing the silver film, as shown in Figure 5, it can not be used multiple times, because that the thin silver film can not be re-used. To avoid the confusions, we delete this sentence from the main text.

Reviewer #2 (Remarks to the Author):

The paper presents an innovative concept for improving contrast in a microscope. However, I miss a comparison with existing systems, such as phase contrast, differential interference contrast, etc.

Response: Thank you very much for your suggestion. I am sorry that we did not present a comparison with existing systems. We have added some descriptions on the comparisons with techniques of phase contrast and DIC in the section Introduction. Please see the following description.

A variety of techniques have been developed to improve image contrast without modification of the samples (label-free imaging), such as phase contrast imaging, differential interference contrast (DIC), Hoffman modulation contrast. These contrast enhancing techniques require specialized and expensive additional components. For examples, the equipment needed for DIC microscopy includes a polarizer, a beam-splitting modified Wollaston prism below the condenser, a beam-recombining modified Wollaston prism above the objective, and an analyzer above this upper prism. The phase contrast and Hoffman modulation contrast techniques need specialized condenser and objective.

中國科學技術大學

UNIVERSITY OF SCIENCE & TECHNOLOGY OF CHINA

These specialized techniques need additional optical or mechanical components, thus complicate the configuration of the microscope and increase the complexities in operations.

I would expect light to be lost through the scattering layer. This still needs to be analyzed.

Response: Thank you for your suggestions.

Yes, there is light to be lost through the scattering layer and the dielectric multilayer. **This is analyzed and additional experiment is carried on, as shown in Figure S11.** For your convenience, we copy the new descriptions in the revised manuscript here.

The light coupling efficiency of the photonic chip was measured with the experimental setup as shown in Figure S11. The light beam from the LED with divergent angles at 3.7° (for 640 nm wavelength) and 10.0° (for 750 nm wavelength) was normal incident onto two control substrates, one is the bare glass substrate, and the other is the designed photonic chip as shown in Figure 1a. An upright objective (NA 1.49, 100X) was used to collect the transmitted light beam whose power was measured with a power meter. For the substrate made of bare glass substrate, the power of the transmitted light is $30.06 \mu\text{W}$ (at 640 nm wavelength) or $9.04 \mu\text{W}$ (at 750 nm wavelength). For the substrate made of the photonic chip, the power of the transmitted light is $8.80 \mu\text{W}$ (at 640 nm wavelength) or $4.5 \mu\text{W}$ (at 750 nm wavelength), then the coupling efficiency can be calculated as 29% (at 640 nm wavelength) or 50% (at 750 nm wavelength). The NA of the objective used to collect the transmitting light is only 1.49, and the transmitted light out of this light-cone cannot be collected, but they also contribute to the darkfield and TIR imaging, so the real coupling efficiency is a little larger than the measured result.

For the conventional darkfield microscopy where an Abbe condenser is used, an opaque spider-style light stop is inserted below the Abbe condenser, the central light rays are blocked, allowing only peripheral light rays to pass through the lenses to form an inverted oblique hollow cone of light. This form of illumination is very wasteful of light and thus demands a high intensity illumination source, meaning that the coupling efficiency of the conventional darkfield microscopy is much low. For the TIR imaging where a high refractive index prism is used to generate the evanescent waves, the coupling efficiency can be as high as 100% if the light path is precisely aligned. However, the precise alignment needs additional components which will add cost and complexity to the microscopy. The prism is also bulk and not suitable for optical integration or compact use. On the contrary, the proposed photonic chip is planar and compact, and the normal incidence does not need specialized component. The coupling efficiency can be further enhanced with optimized parameters of the photonic chip, such as that the narrower angular divergence of the dip (shown in Figure 1d) can induce high coupling efficiency in the case of normal incidence. Due to the planar shape of the photonic chip, in future, the LED light source can be integrated below this chip, then it will be more compact for integration and the coupling efficiency can be further enhanced.

中國科學技術大學

UNIVERSITY OF SCIENCE & TECHNOLOGY OF CHINA

Also, it is not clear to me if the paper is mainly about the photonic chip, or more about a dedicated device for microscopy. The combination of multilayer and surface plasmons has been shown in other work.

Response: Thank you for your comment and reminder. I am sorry that we have not present clearly this point in the original manuscript.

Our paper is mainly about a photonic chip that can work as a dedicated device for microscopy.

The photonic chip is made of multilayer films (either dielectric film or metallic film) and a scattering layer containing nanoparticles. Through placing this photonic chip below the specimens, a standard bright field microscopy (BFM) can be easily transformed to both a chip-based darkfield microscopy (C-DFM) and chip-based total internal reflection microscopy (C-TIRM) without modifying the configuration or the specimens. The principle lies on that, under normal incidence, both the inverted hollow cones of light and various evanescent waves can be generated with this photonic chip due to its tailored angular transmission.

When comparing with bulk Abbe condensers used in a conventional darkfield microscopy (DFM), and prism or oil-immersed objective used in conventional TIR imaging, **this imaging device made of planar chip is more compact, low cost and easy aligned.** It can be fabricated on an extremely large substrate with standard deposition and spin-coating method, without any top-down nanofabrication procedures, thus open new avenues towards the design of a fully integrated on-chip microscopy.

In our paper, we proposed a new design of photonic chip made of two multilayer and a scattering layer for the excitation of both surface plasmons (on metallic film) and evanescent waves (on dielectric medium), which can be used as the illumination source for surface imaging. This proposed photonic chip also can be used to realize the inverted hollow cones of light (Figure 1a) which can be used for darkfield illumination. To the best of our knowledge, we have not seen the similar design.

I think the work has innovative parts that should be published. However, for publication in Nature Communication, the work should focus more on the main innovation. So please emphasize the main innovation.

Response: Thank you very much for your suggestion. I am sorry that I have not emphasized the main innovation. As raised in the first comment by the reviewer, we have not made a comparison with existing systems, such as phase contrast, differential interference contrast, etc, then the innovation is not clearly presented in the original manuscript.

The comparison is added in the revised manuscript, then the innovation of the manuscript is emphasized in both

中國科學技術大學

UNIVERSITY OF SCIENCE & TECHNOLOGY OF CHINA

the section Introduction and Discussion.

For your convenience, the innovation is briefly described as following.

*Modern researchers have proposed many useful techniques to improve the contrast of the optical microscopy images, such as phase contrast imaging, differential interference contrast, darkfield illumination, and Hoffman modulation contrast. **These specialized techniques need additional optical or mechanical components, thus complicate the configuration of the microscope and increase the complexities in operations.***

In this work, a photonic chip made of dielectric multilayer and scattering layer was proposed. We demonstrate that after the attaching of this planar photonic chip to the substrate of a standard brightfield microscopy (BFM), both darkfield and total internal reflection (TIR) imaging can be realized in one experimental setup without the use of a bulky darkfield condenser and other specialized components.

The new microscopes can be named as chip-based darkfield microscopy (C-DFM) and chip-based total internal reflection microscopy (C-TIRM). The C-DFM and C-TIRM have the merits of large illumination area, high imaging contrast, simple configuration and easy for optical-alignment, because that the planar photonic can work under normal incidence (as shown in Figure 1a).

*When comparing with bulk Abbe condensers used in a conventional darkfield microscopy (DFM), and prism or oil-immersed objective used in conventional total internal reflection (TIR) imaging, **this imaging device made of planar chip is more compact, low cost and easy aligned. It can be fabricated on an extremely large substrate with standard deposition and spin-coating method, without any top-down nanofabrication procedures, thus open new avenues towards the design of a fully integrated on-chip microscopy.** Different from the condenser or objective that has limited illumination area (of micrometer scale), this imaging device enables extremely large illumination area up to centimeter-scale or even larger.*

Also a discussion about the coupling efficiency compared to other concepts should be added.

Response: Thank you very much for your suggestion. The coupling efficiency compared to other concepts are added in the revised manuscript, and additional experiments were carried out. as shown in Figure S11. For your convenience, we copy the new discussions in the revised manuscript here.

The light coupling efficiency of the photonic chip was measured with the experimental setup as shown in Figure S11. The light beam from the LED with divergent angles at 3.7° (for 640 nm wavelength) and 10.0° (for 750 nm

wavelength) was normal incident onto two control substrates, one is the bare glass substrate, and the other is the designed photonic chip as shown in Figure 1a. An upright objective (NA 1.49, 100X) was used to collect the transmitted light beam whose power was measured with a power meter. For the substrate made of bare glass substrate, the power of the transmitted light is $30.06 \mu\text{w}$ (at 640 nm wavelength) or $9.04 \mu\text{w}$ (at 750 nm wavelength). For the substrate made of the photonic chip, the power of the transmitted light is $8.80 \mu\text{w}$ (at 640 nm wavelength) or $4.5 \mu\text{w}$ (at 750 nm wavelength), then the coupling efficiency can be calculated as 29% (at 640 nm wavelength) or 50% (at 750 nm wavelength). The NA of the objective used to collect the transmitting light is only 1.49, and the transmitted light out of this light-cone cannot be collected, but they also contribute to the darkfield and TIR imaging, so the real coupling efficiency is a little larger than the measured result.

For the conventional darkfield microscopy where an Abbe condenser is used, an opaque spider-style light stop is inserted below the Abbe condenser, the central light rays are blocked, allowing only peripheral light rays to pass through the lenses to form an inverted oblique hollow cone of light. This form of illumination is very wasteful of light and thus demands a high intensity illumination source, meaning that the coupling efficiency of the conventional darkfield microscopy is much low. For the TIR imaging where a high refractive index prism is used to generate the evanescent waves, the coupling efficiency can be as high as 100% if the light path is precisely aligned. However, the precise alignment needs additional components which will add cost and complexity to the microscopy. The prism is also bulk and not suitable for optical integration or compact use. On the contrary, the proposed photonic chip is planar and compact, and the normal incidence does not need specialized component. The coupling efficiency can be further enhanced with optimized parameters of the photonic chip, such as that the narrower angular divergence of the dip (shown in Figure 1d) can induce high coupling efficiency in the case of normal incidence. Due to the planar shape of the photonic chip, in future, the LED light source can be integrated below this chip, then it will be more compact for integration and the coupling efficiency can be further enhanced.

Reviewer #3 (Remarks to the Author):

The manuscript by Yan Kuai and coworkers describes a compact photonic system to replace the condenser lens issued in dark-field and total internal reflection microscopy (TIRM). Their system is based on a stack of three different planar photonic chips, which are well motivated and characterized in the paper. The first layer system works as an angular filter to collimate the light from the LED source. The second layer is based on dielectric nanoparticles to scatter the light in all directions and the third last layer is again an angular filter. Depending on the illumination wavelength, the light finally transmitted to the sample occurs at high angles (for dark-field microscopy at 750 nm) or corresponds to evanescent waves (for TIRM at 640 nm). I believe this approach is quite original and I am unaware

中國科學技術大學

UNIVERSITY OF SCIENCE & TECHNOLOGY OF CHINA

of anything similar.

Altogether, while I find the concept original and interesting, I somehow miss a valuable groundbreaking application here. The authors essentially show images of dielectric nanowires and 200 and 50 nm polystyrene nanoparticles in air, thereby taking advantage of the high refractive index contrast between the polystyrene and air. That makes me seriously question the potential impact of this work on the microscopy community which is more likely to look for samples immersed into water.

Response: Thank you very much for your suggestion. Yes, it will have high potential impact if we can image the samples immersed into water.

I am sorry that I have not given a more clearly or directly description on this point in the original manuscript. We have shown that the proposed method can work when the specimens (polymer microwires and nanowires) are immersed in water solution, as shown in Figure S9.

To strengthen this point, we also image the live biological cells immersed into liquid solution using the proposed method, as shown in the Figures S10 of the revised manuscript.

The paper is nicely written but honestly, I find this is currently more the type of work found in ACS Photonics and related journals. I would like to recommend publication in Nature Communications, but then I would require the authors to show a more compelling demonstration of the merit of their system on a more challenging sample for imaging. For instance, working on a biological cell sample in water medium and using TIRM eventually combined to fluorescence to image the basal cell membrane only. This is what TIRFM is typically used for. An even more potentially impressive application would go to UV illumination of fluorescence, where the choice of optical condenser elements is much more restricted (but UV-LED with quite high power are available). However, this implies changing completely the wavelength range used by the authors so far, and I would understand if the authors prefer to stay in the red spectral they are familiar with.

Response: Thank you very much for your suggestion and we really appreciate the wonderful ideas related with our work. Based on your suggestions, we have tried a more challenging samples for imaging, such as the polymer wires in water solution, and live biological cells in culture solution. Experimental results shown in Figures S9, and S10 prove the feasibility of our system. The images of the nanowire (Figure S9) appear discontinuous when it crosses the microwire (Figure S9c) verify that our system can image the specimens attached to the surface only. In the cross region of the polymer nanowire and microwire, the nanowire is not attached on the surface of the photonic chip, so this region is not be imaged. The images (Figure S10) of the live cells show more distinct boundaries of the cells, demonstrating the typical characteristic of darkfield and TIR imaging.

中國科學技術大學

UNIVERSITY OF SCIENCE & TECHNOLOGY OF CHINA

In our work, we mainly focus on the label-free imaging, so the samples are all non-fluorescent, and we do not capture the fluorescence images. The typical total internal reflection fluorescence microscopy is always used for fluorescence image, which is beyond the main focus of our current work.

I understand that it is very important to shift the working wavelengths to the UV band, where the choice of conventional optical condenser elements is more much restricted, then the innovations of the proposed photonic chip for imaging (for both fluorescence imaging and label-free imaging) will be more obvious and attractive. I think this suggestion will result in another novel paper in this field, which will be carried out in future. We thank the reviewer for give us this wonderful idea.

Other comments:

- Many sub-figures lack a description of the quantity plotted (Fig. 1b&c, 2c&d, 3d&f...). Sometimes it is the reflectivity, sometimes the transmission. This should be clarified every time.

Response: Thank you very much for your reminder. I am sorry for this confusion. We have clarified the description of the quantity in the caption of Figure 1, Figure 2 and Figure 3.

- I miss to see a description of the numerical simulation methods and approach

Response: Thank you very much for your reminder. Our simulation methods and approach are based on the transfer matrix methods, which is a typical method for numerically calculating the reflectivity of layer structures. And the details of this method are described in References 20. Please see the first paragraph of the section Results.

- I would like to see a comparison between experiments (Fig. 2c, d) and simulations (Fig. 1b, c) for the two selected wavelengths 640 & 750 nm. Why not remove Fig. 1b, c to the SI and add the experimental characterization data at the bottom of Fig. 1 in the same fashion as Fig. 1d, e? I believe this would convince the reader more easily.

Response: Thank you very much for your suggestions. I totally understand that your suggestion will present a different way to demonstrate the optical properties of the proposed photonic chip. In our original manuscript, Figure 1b and 1c show the optical properties of the two dielectric-multilayers at different incident wavelengths, and I think many readers will be curious on this point, so we put these results in the main text.

According to this suggestion, we put the experimental reflection curves at the two selected wavelengths 640 & 750 nm at the bottom of the revised Figure 2 (Figures 2e and 2f). On Figure 2e, the reflection dips appear at the normal incidence (incident angle near 0° , both TM and TE polarization) for the bottom multilayer, which are consistent with

中國科學技術大學

UNIVERSITY OF SCIENCE & TECHNOLOGY OF CHINA

dips shown in the simulated curves (Figure 1d). On Figure 2f, there are dips appear in the case of TM polarized incident and incident wavelength at 750 nm, and the position of dips is consistent with that shown in Figure 1e (simulated curves).

The slightly difference between the experimental and simulated curves is that the reflectivity for the TE-polarized light gradually decreases with the increasing incident angle, which can be attributed to the depolarization effect of the high NA objective used in the BFP imaging (redacted).

We have added these new descriptions in the revised manuscript.

- line 459, "Light filed...", do the authors mean "light field"?

Response: Thank you very much for your reminder. Yes, we mean "light field", and I am sorry for my typos.

- Fig. 3e&g, I do not understand the vertical scaling and why it is only a few percent. A few percent of what reference? That leads to the question: how much light is lost by this condenser? What is the available power transmitted to the sample?

Response: Thank you very much for your reminder.

Figures 4e and 4f are to show that most of the transmitted light is of large transmitting angle (form the hollow cones of light as shown in Figure 1a), so that the transmitting light will be out of the collection angle of the imaging objective (NA = 0.7 or 0.6 here). As a result, darkfield and TIR imaging can be realized.

The ratio of the transmitted light out of the NA of the imaging objective to the total transmitted light is vital for the contrast of the darkfield imaging.

On Figures 3e and 3g, the horizontal axis is the NA that is defined by diameter of the ring labelled on the back focal plane (BFP) images in Figures 3d and 3f. One ring corresponds to one NA. **The vertical axis is the intensity ratio of the transmitted light on the corresponding ring (each NA) to the total transmitted light on the BFP images.**

Figure 3e and Figure 3g show that more than 98% of the total transmitted light is out of the collection angle of the imaging objective (defined by the NA = 0.7), which guarantee the high contrast of the following darkfield images.

中國科學技術大學

UNIVERSITY OF SCIENCE & TECHNOLOGY OF CHINA

To make sure how much light is lost by this condenser and the available power transmitted to the sample, **we did additional experiments**, as shown in Figure S11. For your convenience, we copy the new descriptions in the revised manuscript here.

The light coupling efficiency of the photonic chip was measured with the experimental setup as shown in Figure S11. The light beam from the LED with divergent angles at 3.7° (for 640 nm wavelength) and 10.0° (for 750 nm wavelength) was normal incident onto two control substrates, one is the bare glass substrate, and the other is the designed photonic chip as shown in Figure 1a. An upright objective (NA 1.49, 100X) was used to collect the transmitted light beam whose power was measured with a power meter. For the substrate made of bare glass substrate, the power of the transmitted light is $30.06 \mu\text{w}$ (at 640 nm wavelength) or $9.04 \mu\text{w}$ (at 750 nm wavelength). For the substrate made of the photonic chip, the power of the transmitted light is $8.80 \mu\text{w}$ (at 640 nm wavelength) or $4.5 \mu\text{w}$ (at 750 nm wavelength), then the coupling efficiency can be calculated as 29% (at 640 nm wavelength) or 50% (at 750 nm wavelength). The NA of the objective used to collect the transmitting light is only 1.49, and the transmitted light out of this light-cone cannot be collected, but they also contribute to the darkfield and TIR imaging, so the real coupling efficiency is a little larger than the measured result.

For the conventional darkfield microscopy where an Abbe condenser is used, an opaque spider-style light stop is inserted below the Abbe condenser, the central light rays are blocked, allowing only peripheral light rays to pass through the lenses to form an inverted oblique hollow cone of light. This form of illumination is very wasteful of light and thus demands a high intensity illumination source, meaning that the coupling efficiency of the conventional darkfield microscopy is much low. For the TIR imaging where a high refractive index prism is used to generate the evanescent waves, the coupling efficiency can be as high as 100% if the light path is precisely aligned. However, the precise alignment needs additional components which will add cost and complexity to the microscopy. The prism is also bulk and not suitable for optical integration or compact use. On the contrary, the proposed photonic chip is planar and compact, and the normal incidence does not need specialized component. The coupling efficiency can be further enhanced with optimized parameters of the photonic chip, such as that the narrower angular divergence of the dip (shown in Figure 1d) can induce high coupling efficiency in the case of normal incidence. Due to the planar shape of the photonic chip, in future, the LED light source can be integrated below this chip, then it will be more compact for integration and the coupling efficiency can be further enhanced.

- I am not a great fan of Figure 5. I understand the idea to show this system is also able to generate surface plasmon waves on a thin silver interface, but I find that (1) this is quite obvious and expected and (2) this is introducing a different system into the story, so that the reader then asks the question which device is the best and what is the interest for the other one then?

Response: Thank you very much for your suggestion.

中國科學技術大學

UNIVERSITY OF SCIENCE & TECHNOLOGY OF CHINA

Yes, it is quite obvious and expected that this system is able to generate surface plasmons waves on a thin silver interface. Different from the dielectric film used in Figure 4, the silver film is conductive, then the proposed experimental configuration has the potential to work as an electrochemical microscopy for imaging local electrochemical current, studying heterogeneous surface reactions and for analyzing trace chemicals. The extension of proposed photonic chip from all-dielectric configuration to dielectric-metal hybrid configuration will expand the potential applications of the proposed photonic chips, and show the universality of the proposed method for generating various surface waves (either surface plasmons waves or evanescent waves).

However, metallic film, such as the silver film, is not stable in air due to the oxidation. It is not stable and can not be used for long time. On the contrary, the dielectric film is much more stable and can be used for many times. It is difficult to say which device is absolutely better than the other. They have different applications.

On the other hand, now there are many researchers working on surface imaging by using surface plasmon waves. The extension of our system to silver interface will make the paper more attractive to broad readers.

Based on the above thoughts, we show that this system is also able to generate surface plasmon waves on a thin silver interface.

List of Changes

1. As suggested by Reviewer 2, We have added some descriptions on the comparisons with techniques of phase contrast and DIC in the section Introduction.
2. As suggested by Reviewer 3, we have added new experimental results on imaging the samples immersed into water, such as imaging the live biological cells in liquid solution. The experimental results are shown in Figure S10 in the supplementary information.
3. We have added the sections on data availability and code availability.
4. We have corrected the typos in the main text, supplementary information and Figures, which are raised by Reviewer 1. The manuscript is also edited by native English speaker, Prof Lakowicz.
5. As suggested by Reviewer 1, we replace the word “amplify” with “enhance”.
6. As suggested by Reviewer 1, we rephrase the sentences in Lines 140-141, Line 156-157, line 160-162, Line 184-

中國科學技術大學

UNIVERSITY OF SCIENCE & TECHNOLOGY OF CHINA

185, of the original manuscript.

7. As suggested by Reviewer 1, we added a reference for the description on Line 205, the references 27.
8. As reminded by Reviewer 1, we delete the sentence in Line 222-223 to avoid the confusions.
9. As suggested by Reviewer 2 and Reviewer 3, we analyze the coupling efficiency of the light to the photonic chip.
10. As suggested by Reviewer 2, we emphasize the main innovation of our work.
11. As suggested by Reviewer 3, we have clarified the description of the quantity in the caption of Figure 1, Figure 2 and Figure 3.
12. As suggested by Reviewer 2, we add two new images on Figure 2, to made a good comparison between the numerical and simulated results.
13. As suggested by Reviewer 3, we correct the typos on Line 459 of the original manuscript.
14. As reminded by Reviewer 3, we give some addition descripts on why we carried out the experiment shown Figure 5.
15. I have updated our current address and the acknowledgement.
16. We have added Mr. Zetao Fan as the co-author. He has contributed to the imaging of biological cells in water, and also the measurement of the coupling efficiency.

REVIEWERS' COMMENTS

Reviewer #1 (Remarks to the Author):

Authors have considered my and other reviewer's remarks and I am satisfied by their answers. Therefore, I recommend the article to be accepted for publication.

Reviewer #2 (Remarks to the Author):

The paper has been revised according to the suggestions of the reviewers.

Reviewer #3 (Remarks to the Author):

My main comment concerned the application to living cells. The revised Figure S10 performed such experiment, in a very minimalist way. While this is demonstrating the concept, the image remains of relatively poor quality.

Concerning the rest of the manuscript, the revisions have been performed in an appropriate manner.

IMPORTANT NOTICE: the revised figure 1 is not correct, it is a duplicate of figure 4. The authors must correct this and upload their previous figure 1.

A Point-by-point response to the referees' comments

REVIEWERS' COMMENTS

Reviewer #1 (Remarks to the Author):

Authors have considered my and other reviewer's remarks and I am satisfied by their answers. Therefore, I recommend the article to be accepted for publication.

Response: Thank you for your kind comment.

Reviewer #2 (Remarks to the Author):

The paper has been revised according to the suggestions of the reviewers.

Response: Thank you for your kind comment.

Reviewer #3 (Remarks to the Author):

My main comment concerned the application to living cells. The revised Figure S10 performed such experiment, in a very minimalist way. While this is demonstrating the concept, the image remains of relatively poor quality. Concerning the rest of the manuscript, the revisions have been performed in an appropriate manner.

Response: Thank you for your kind comment. For application to living cells, the cells are live and not immobilized to the substrate, so their images are not as good as those of the polymer wires or nanoparticles. The present images clearly verify the ability of the proposed method for live cells. In future, for practical application in biological cells' imaging, the proposed photonic chip can be modified to be more compatible with the live cells, then the images quality can be highly improved.

IMPORTANT NOTICE: the revised figure 1 is not correct, it is a duplicate of figure 4. The authors must correct this and upload their previous figure 1.

Response: Thank you for your kind reminder. I apologize for my mistake.

I use the right Figure 1 in the revised version. Thank you again for your reminder.

A list of changes

1. We have replaced the wrong Figure 1 with the right one, as reminded by Reviewer 3.
2. Any abbreviations, symbols or colours present in the figures have been defined in the associated legends.

中國科學技術大學

UNIVERSITY OF SCIENCE & TECHNOLOGY OF CHINA